

# Association between free thyroxine levels and clinical phenotype in first-episode psychosis: a prospective observational study

Eloi Gine-Serven[1], Maria Martinez-Ramirez[1], Ester Boix-Quintana[1], Eva Davi-Loscos[1], Nicolau Guanyabens[2], Virginia Casado[2], Desiree Muriana[2], Cristina Torres-Rivas[1], M.J. Cuesta[3,4] and Javier Labad[1,5,6]

[1] Department of Mental Health, Hospital de Mataró, Consorci Sanitari del Maresme, Mataró, Spain
[2] Department of Neurology, Hospital de Mataró, Consorci Sanitari del Maresme, Mataró, Spain
[3] Department of Psychiatry, Complejo Hospitalario de Navarra, Pamplona, Spain
[4] IdiSNA, Navarra Institute for Health Research, Pamplona, Spain
[5] Translational Neuroscience Research Unit I3PT-INc-UAB, Institut de Innovació i Investigació Parc Taulí (I3PT), Institut de Neurociències, Universitat Autònoma de Barcelona, Barcelona, Spain
[6] Centro de Investigación en Red de Salud Mental (CIBERSAM), Madrid, Spain

Corresponding author
Eloi Gine-Serven, elogi69@gmail.com

## ABSTRACT

**Aim.** To determine whether thyroid hormone levels are associated with a specific clinical phenotype in patients with first-episode psychosis (FEP).

**Methods.** Ninety-eight inpatients experiencing FEP and with less than 6 weeks of antipsychotic treatment were included in the study and were followed up for one year. Baseline psychiatric evaluation included assessment of prodromal symptoms, positive and negative symptoms, depressive symptoms, stressful life events and cycloid psychosis criteria. Thyroid function (thyroid-stimulating hormone (TSH) and free thyroxin (FT4)) was determined at admission. Partial correlation analysis was conducted to analyse the correlation between levels of TSH/FT4 and symptoms. Logistic regression was performed to explore the association between psychopathological symptoms, 12-month diagnoses and thyroid hormones while adjusting for covariates.

**Results.** Patients with prodromal symptomatology showed lower baseline FT4 levels (OR = 0.06; $p = 0.018$). The duration of untreated psychosis (DUP) was inversely associated with FT4 concentrations ($r = -0.243$; $p = 0.039$). FEP patients with sudden onset of psychotic symptoms (criteria B, cycloid psychosis) showed higher FT4 levels at admission (OR = 10.49; $p = 0.040$). Patients diagnosed with affective psychotic disorders (BD or MDD) at the 12-month follow-up showed higher FT4 levels at admission than patients diagnosed with nonaffective psychosis (schizophrenia, schizoaffective) (OR = 8.57; $p = 0.042$).

**Conclusions.** Our study suggests that higher free-thyroxine levels are associated with a specific clinical phenotype of FEP patients (fewer prodromal symptoms, shorter DUP duration and sudden onset of psychosis) and with affective psychosis diagnoses at the 12-month follow-up.

## INTRODUCTION

First-episode psychosis (FEP) refers to heterogeneous clinical conditions representing the symptomatic emergence of myriad disorders, for instance, schizophrenia, schizoaffective disorder or bipolar disorder (*Giné-Servén et al., 2022*); with a pooled median point and 12-month prevalence of approximately four persons per 1,000 (*Moreno-Küstner, Martín & Pastor, 2018*). Such disorders are predominantly preceded by a prodromal phase, commonly lasting months or years, in which faint symptoms show and are concurrent with a reduction of functionality in different areas, including sociofamiliar relationships or academic and occupational performance (*Woodberry et al., 2016*). This prodromal phase, as well as the duration of untreated psychosis (DUP) and the duration of untreated illness (DUI), is persistently related to functional recovery in patients with FEP (*Santesteban-Echarri et al., 2017*).

Not yet completely understood are the aetiology and pathogenesis of psychosis; however, the overwhelming evidence points to a contribution from a combination of genetic and environmental factors (*Brown, 2011*; *Tsuang et al., 2004*). The lack of biomarkers adds to diagnostic delay as well as obstructing disease stratification, prediction of outcomes and therapeutic choice (*Weickert et al., 2013*).

Some studies have indicated that hormone deregulation may play a role in the development of psychosis (*Hayes, Gavrilidis & Kulkarni, 2012*). Patients with schizophrenia usually develop endocrine abnormalities, such as hyperprolactinemia (*Labad, 2019*; *González-Blanco et al., 2016*), dysfunction of the hypothalamic–pituitary–adrenal (HPA) axis or release of neurosteroids and appetite-regulating hormones (*Misiak et al., 2021*). Individual-level risk factors, such as poor dietary habits, sedentary behaviour and adverse effects of antipsychotics, also determine whether these patients will develop endocrine abnormalities (*Misiak et al., 2021*). Nevertheless, some groups (*González-Blanco et al., 2016*; *Hubbard & Miller, 2019*; *Lis et al., 2020*) have hypothesized that hyperprolactinemia, insulin resistance and HPA axis alterations might be related to intrinsic pathophysiological mechanisms and could also occur in early psychosis.

Thyroid hormones could very well play vital roles in the development and correct function of the CNS, supporting the development of neurons, oligodendrocytes, astrocytes and microglia and additionally, the modulation of proinflammatory feedback (*Noda, 2015*). Since neuroinflammation is said to have been associated with the pathogenesis of schizophrenia (*Howes & McCutcheon, 2017*), thyroid hormones might also assist in the pathogenesis and clinical expression of schizophrenia and psychotic disorders by virtue of a proinflammatory mechanism.

Altered hypothalamic-pituitary-thyroid system function has been described in schizophrenia and bipolar and depressive disorders (*Bicikova et al., 2011*; *Othman et al., 1994*; *Santos et al., 2012*), but very little research has been done in the early stages of these psychiatric disorders. In cross-sectional (*Barbero et al., 2015*) and prospective (*Labad et al., 2016*) studies carried out by our team, although within the normal range, higher free thyroxine (FT4) levels were associated with better attention and vigilance in early-stage psychotic disorders. Higher FT4 levels have been described in patients with affective

psychosis than in those with nonaffective psychosis (*Barbero et al., 2015*). No associations were found by another group between thyroid function and positive, negative or general symptoms in drug-naïve male patients with schizophrenia (*Jose et al., 2015*).

However, few studies have explored the relationship between the clinical expression and thyroid hormones in first-episode psychosis (FEP). Previously carried out investigation highlights the necessity to perform research on biomarkers in the early stages of psychosis, which, it is thought, could provide crucial clues to the mechanisms underlying psychotic disorders and in turn, permitting the minimalization of confounders, in particular, antipsychotic exposure and the neurodegenerative evolution of the disease.

In the current study, we aimed to determine if thyroid hormones in patients with FEP related to a distinct 12-month follow-up diagnoses and a differentiated clinical phenotype. We hypothesized that thyroid hormone levels were associated with clinical diagnosis one year after the onset of psychosis, with higher FT4 concentrations in affective psychoses than in nonaffective psychoses. As a secondary aim, we explored whether thyroid hormone levels were associated with distinct phenotypes (sudden onset, severity) in FEP patients.

## METHODS

Data were collected as previously described in *Giné-Servén et al. (2022)*.

### Study design and participants

Ninety-eight patients experiencing FEP were included in the study. These patients were admitted to acute inpatient units (adult or child and adolescent units) from the Department of Mental Health at the Hospital of Mataró between 1 June 2018 and 31 March 2020. FEP was defined as new-onset disorganized behaviour accompanied by delusions or hallucinations not caused by drugs that met DSM-IV criteria for a psychotic disorder (schizophrenia, bipolar disorder or unipolar major depression with psychotic features, schizophreniform disorder, brief psychotic disorder, delusional disorder, psychotic not otherwise specified). Patients were excluded if they had (1) positive symptoms of psychosis lasting more than 6 months; (2) treatment with antipsychotics, antidepressants or mood stabilizers for more than 6 weeks; (3) a past history of positive symptoms of psychosis; (4) a previous diagnosis of intellectual disability (IQ < 70), or (5) active medical or neurological diseases that could explain the current symptoms.

The study received approval from the local ethics committees (Hospital of Mataró, Barcelona, Spain; Institute Ethics Committee number: *1/18*). All participants were informed about the nature of the study and gave written informed consent for participating in the study.

### Clinical assessment

During the first week of hospital admission, all patients underwent psychiatric and neurological evaluations. The description of the clinical and biological assessment is described in Figure 1 of the supplementary material (Fig. S1). Two trained attending psychiatrists carried out diagnostic interviews using the Structured Clinical Interview for DSM-IV-TR (SCID-I) (*First, Spitzer & Gibbon, 1994*) for patients ≥18 years, the Schedule

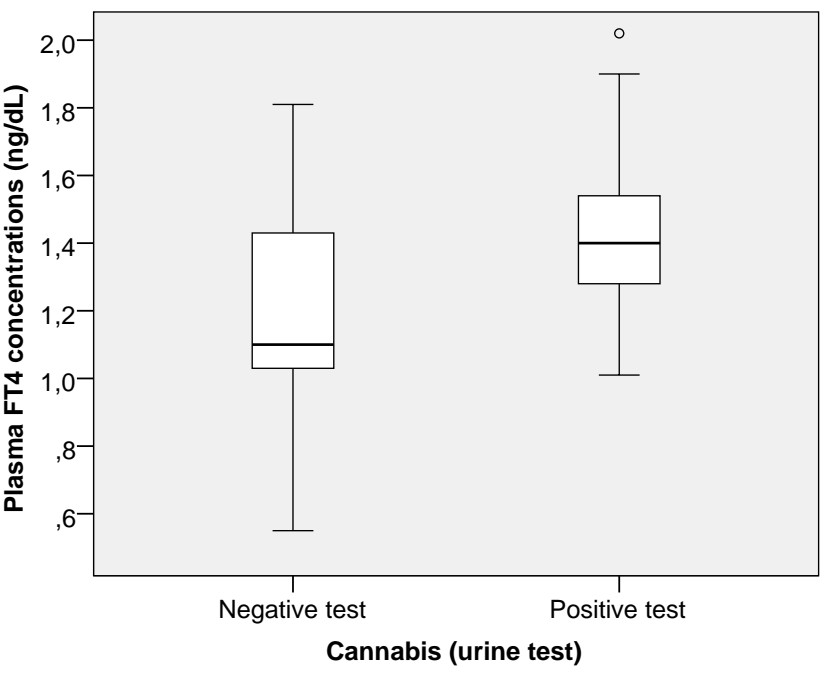

**Figure 1** Cannabis FT4.

for Affective Disorders and Schizophrenia for school-age children and the Present and Lifetime version (K-SADS-PL) (*Kaufman et al., 1997*) for patients <18 years.

The onset of prodromal and psychotic symptoms was assessed retrospectively by means of a semistructured interview with a specific *ad hoc* inventory (Quick Psychosis Onset and Prodromal Symptoms Inventory (Q-POPSI)) that was designed for administration to patients and family and/or close relatives (*Giné-Servén et al., 2022*). The DUI and DUP were calculated. DUI was defined as the difference in time between the onset of the first symptom (prodromal or psychotic) of the illness and the start of the antipsychotic treatment. DUP was defined as the difference in time between the onset of the first positive psychotic symptom and the start of antipsychotic treatment. A full explanation of the Q-POPSI inventory is described elsewhere (*Giné-Servén et al., 2022*).

Psychopathology at admission was assessed using three psychometric scales. The Positive and Negative Syndrome Scale (PANSS) (*Kay, Fiszbein & Opler, 1987*) was used to assess positive, negative and general psychopathology symptoms. Symptoms were recoded into five subscales following the *Wallwork et al. (2012)* consensus: positive, negative, disorganized/concrete, excited and depressed factors. Acute psychosis onset was assessed with cycloid psychosis criteria (Table S1: Perris and Brockington's diagnostic criteria for cycloid psychosis) (*Brockington et al., 1982; Perris, 1974*). The Young Mania Rating Scale (YMRS) (*Lukasiewicz et al., 2013; Young et al., 1978*) was administered to assess manic symptoms. The Hamilton Depression Rating Scale (HAM-D) (*Hamilton, 1960; Zimmerman et al., 2013*) was also administered to assess depressive symptoms.

Stressful life events that occurred during the 6 months prior to admission were assessed using The List of Threatening Experiences (*Brugha & Cragg, 1990*), a subset of 12 life event categories that are associated with considerable long-term contextual threat.

Functional outcome was assessed at admission and discharge from the Acute Inpatient Unit using the Global Assessment of Functioning Scale (GAF) (*Hall, 1995*) for patients ≥18 years and the Children's Global Assessment Scale (C-GAS) (*Shaffer, 1983*) for patients < 18 years.

## Thyroid function studies

During the first 24–48 h of admission, a fasted morning (between 8:30 and 9:30 AM) blood sample was obtained to determine thyroid hormone concentrations. Levels of TSH and FT4 were determined under routine conditions on the same day of the blood draw. The hormonal assay was performed in the Hospital de Mataró Testing Laboratory. TSH was determined by electrochemiluminescence immunoassay (ECLIA); levels between 0.30–4.20 mcUI/mL were considered normal for both sexes. Free thyroxine (FT4) was evaluated by chemiluminescence (ECLIA); levels between 0.80−1.84 ng/dL were considered normal for both sexes (Cobas 8000 e801 (Roche, Basel, Switzerland)). FT4 and TSH inter- and intra-assay coefficients of variability (CV) were < 5%. TSH concentrations were determined in all patients, but FT4 was available in only 70 out of 98 patients.

## Statistical analysis

All data analyses were performed using IBM SPSS Statistics for Windows, Version 20.0 (IBM Corporation, Armonk, NY, USA). Partial correlation analyses were used to explore the correlation between TSH and FT4 concentrations and continuous measures (DUI, DUP and psychometric scores) while adjusting for age and sex. Significance was set as a $p$ value < 0.05 (bilateral).

Logistic regression was performed to explore the association between thyroid hormones (TSH, FT4) and affective psychosis while adjusting for sex and age. In this analysis, affective psychosis (defined as a diagnosis of bipolar disorder or psychotic depression confirmed at the follow-up visit at one year) was used as the dependent variable. The reference category for psychosis diagnosis was nonaffective psychosis (schizophrenia-spectrum diagnoses).

Further exploratory analyses were conducted to study the association between thyroid hormones and other clinical variables dealing with the presentation of first-episode psychosis (presence of prodromal symptoms; cycloid psychosis criteria) while adjusting for covariates. In these analyses, sex, age, cannabis use (defined as a positive cannabis test at admission) and stressful life events at onset were considered independent variables.

## RESULTS

The demographic, clinical and biochemical data of the sample at the baseline assessment are described in Table 1. Raw patient data used for the statistical analysis is provided as (Table S2). Fifty-five FEP patients (56.1%) had prodromal symptoms. Thirty FEP patients (30.6%) had sudden onset of psychotic symptoms (criteria B, cycloid psychosis). Nine FEP patients (9.2%) met the full criteria for cycloid psychosis. Regarding thyroid hormone

concentrations, the mean values for TSH and FT4 concentrations were within the normal limits. The ranges for TSH concentrations were 0.49 to 8.18 mcUI/mL and 0.55 to 2.02 ng/dL for FT4. When the blood tests to determine thyroid function were performed, none of the patients were under lithium medication.

### Clinical phenotype and free thyroxine levels at onset (exploratory analyses)

Partial correlations (adjusted for age and sex) showed a negative significant association between TSH and FT4 concentrations (r = −0.355, $p = 0.002$). FT4 was negatively associated with DUP (r = −0.243, $p = 0.039$) but not DUI. TSH and FT4 concentrations were not associated with the severity of psychopathological scales, such as PANSS positive, negative, general psychopathology subscores, PANSS total score, YMRS or HAM-D (data not shown).

Patients with a positive cannabis test in urine at admission had higher FT4 concentrations (1.43 ± 0.25 *vs.* 1.21 ± 0.29, $p = 0.002$; Fig. 1), although they had similar TSH concentrations (1.73 ± 1.49 *vs.* 1.91 ± 1.07, $p = 0.500$).

The results of the logistic regression exploring the relationship between thyroid hormones and the clinical phenotype at the onset of psychosis are described in Tables 2–4. FT4 (but not TSH) concentrations were associated with prodromal symptoms at the onset (OR = 0.06, $p = 0.018$) and B criteria of cycloid psychosis (sudden onset; OR = 10.49, $p = 0.040$). Other cycloid psychosis criteria (A, C, D and full criteria) were not associated with thyroid function.

### Diagnosis at the 12-month follow-up and free thyroxine levels at onset (main hypothesis)

Fifty-four FEP patients (55.1%) were diagnosed with affective psychotic disorders (BD or MDD) at the 12-month follow-up. In the logistic regression analysis exploring the relationship between baseline thyroid hormone concentrations at admission and the diagnosis at follow-up (1 year) and adjusted for age and sex, FT4 concentrations were associated with affective psychosis (OR = 8.57, $p = 0.042$). TSH concentrations were not associated with the clinical diagnosis (OR = 1.26, $p = 0.273$).

## DISCUSSION

In our study, which included 98 FEP patients, higher (but in normal range) plasma FT4 concentrations at admission were associated with a specific clinical phenotype, characterized by fewer prodromal symptoms, shorter DUP duration and a sudden onset of psychosis. Cannabis use was associated with higher FT4 concentrations. Regarding the comparison between affective and nonaffective psychosis, patients diagnosed with affective psychosis (bipolar disorder or psychotic depression) at the 12-month follow-up showed higher free thyroxine levels at admission.

Higher free thyroxine was associated with sudden onset of psychosis, as assessed with Perris & Brockington cycloid psychosis criteria (Perris, 1974). Thyrotoxicosis and autoimmune thyroiditis have been associated with acute and severe episode

**Table 1  Demographic, clinical and biochemical variables of 98 patients with first-episode psychosis.**

| | |
|---|---|
| Age, mean (SD), years | 34.7 (15.3) |
| Female sex, n (%) | 40 (40.8%) |
| Previous history of psychiatric (nonpsychotic) disorders, n (%) | 60 (61.9%) |
|     Mood disorder | 38 (38.7%) |
|     Anxiety disorder | 7 (7.1%) |
|     Obsessive compulsive disorder | 2 (2.0%) |
|     Personality disorder | 2 (2.0%) |
|     Eating behaviour disorder | 4 (4.1%) |
|     Substance use disorder | 28 (28.6%) |
|     Others | 7 (7.1%) |
| Smoking, n (%) | 51 (52.0%) |
| Cannabis use (abuse or dependence), n (%) | 48 (49.0%) |
| Alcohol use (abuse or dependence), n (%) | 29 (29.6%) |
| First degree family history of psychiatric disease, n (%) | 44 (44.9%) |
| Previous life stressful events, n (%) | 41 (41.8%) |
| Prodromal symptoms, n (%) | 55 (56.1) |
|     Cognitive symptoms | 22 (22.4%) |
|     Negative symptoms | 23 (23.5%) |
|     Attenuated positive psychotic symptoms | 28 (28.6%) |
|     Mood symptoms | 21 (21.4%) |
|     Anxiety symptoms | 24 (24.5%) |
|     Obsessive-compulsive symptoms | 4 (4.1%) |
| Duration of psychiatric prodromal symptoms, n (%) | |
|     No prodromal symptoms | 43 (43.9%) |
|     <1 month | 4 (4.1%) |
|     1–6 months | 24 (24.5%) |
|     >6 months | 27 (27.6%) |
| Duration of untreated illness, mean (SD), days | 173.2 (257.1) |
| Duration of untreated psychosis, mean (SD), days | 37.5 (50.9) |
| Treatment during hospital admission, n (%) | |
|     Atypical antipsychotics | 98 (100%) |
|     Typical antipsychotics | 8 (8.2%) |
|     Mood stabilizers | 48 (49.0%) |
|     Electroconvulsive therapy | 1 (1.0%) |
| PANSS, mean (SD) | |
|     Total score | 82.3 (20.0) |
|     Wallwork factors: | |
|         Positive factor | 14.4 (3.5) |
|         Negative factor | 12.1 (7.0) |
|         Disorganized/concrete factor | 8.8 (2.9) |
|         Excited factor | 11.6 (4.1) |

**Table 1** (*continued*)

| | |
|---|---|
| Depressed factor | 8.2 (3.6) |
| YMRS, mean (SD) | 26.7 (11.2) |
| HAM-D, mean (SD) | 22.5 (9.7) |
| Cycloid psychosis complete phenotype, n (%) | 9 (9.2%) |
| Acute psychotic episode | 36 (36.7%) |
| Sudden onset | 30 (30.6%) |
| Clinical profile (4 or more) | 23 (23.5%) |
| Confusion | 27 (27.6%) |
| Delusions | 98 (100%) |
| Hallucinations | 53 (54.1%) |
| Pananxiety | 24 (24.5%) |
| Happiness or ecstasy | 40 (40.8%) |
| Motility disturbances | 25 (25.5%) |
| Concern with death | 8 (8.2%) |
| Oscillations of mood | 10 (10.2%) |
| No fixed symptomatologic combination | 11 (11.2%) |
| Suicidal attempt during FEP | 9 (9.2%) |
| GAF, mean (SD) | |
| On admission | 29.9 (6.6) |
| At discharge | 62.5 (9.1) |
| Thyroid hormones levels | |
| Parameters, mean (SD) | |
| TSH (mcUI/mL) | 1.8 (1.2) |
| FT4 (ng/dL) | 1.3 (0.2) |

**Notes.**

Abbreviations: SD, standard deviation; PANSS, positive and negative syndrome scale; YMRS, Young Mania Rating Scale; HAM-D, Hamilton Depressive Rating Scale for Depression; GAF, Global Assessment of Functioning; FEP, first-episode psychosis; TSH, thyroid stimulating hormone; FT4, free thyroxine.

**Table 2** Results of the logistic regression exploring the relationship between thyroid hormones and clinical phenotype at onset.

| | Prodromal symptoms at onset | | | Full criteria cycloid psychosis | | |
|---|---|---|---|---|---|---|
| | OR | CI 95% OR | *p* value | OR | CI 95% OR | *p* value |
| TSH | 1.026 | 0.678–1.553 | 0.903 | 0.925 | 0.436–1.963 | 0.840 |
| FT4 | 0.059 | 0.006–0.621 | 0.018 | 1.779 | 0.082–38.638 | 0.714 |
| Female gender | 0.373 | 0.111–1.248 | 0.11 | 1.635 | 0.313–8.553 | 0.560 |
| Age | 0.921 | 0.877–0.968 | 0.001 | 0.950 | 0.884–1.020 | 0.158 |
| SLE at onset | 3.338 | 0.935–11.922 | 0.063 | 1.575 | 0.281–8.833 | 0.606 |
| Cannabis (positive urine test) | 1.676 | 0.470–5.970 | 0.426 | 0.355 | 0.52–2.413 | 0.289 |
| Nagelkerke $R^2$ of the model | 0.344 | | | 0.118 | | |

**Notes.**

Abbreviations: TSH, thyroid stimulating hormone; FT4, free thyroxine; SLE, stressful life events.

psychotic induction (*Brownlie et al., 2000*; *Menon, Subramanian & Thamizh, 2017*). Possible mechanisms could involve excess thyroid hormone affecting neurotransmitters (serotonin, gaminobutyric acid and dopamine) or second messengers (adenyl cyclase

**Table 3  Results of the logistic regression exploring the relationship between thyroid hormones and clinical phenotype at onset.**

| | Criteria A cycloid psychosis | | | Criteria B cycloid psychosis | | |
|---|---|---|---|---|---|---|
| | OR | CI 95% OR | p value | OR | CI 95% OR | p value |
| TSH | 1.114 | 0.758–1.637 | 0.584 | 1.440 | 0.949–2.183 | 0.086 |
| FT4 | 2.252 | 0.301–16.844 | 0.428 | 10.485 | 1.109–99.096 | 0.040 |
| Female gender | 1.057 | 0.386–2.895 | 0.914 | 0.899 | 0.299–2.705 | 0.849 |
| Age | 0.989 | 0.953–1.026 | 0.552 | 0.976 | 0.937–1.017 | 0.251 |
| SLE at onset | 1.478 | 0.531–4.112 | 0.454 | 1.863 | 0.603–5.756 | 0.279 |
| Cannabis (positive urine test) | 0.613 | 0.194–1.942 | 0.406 | 0.339 | 0.92–1.251 | 0.104 |
| Nagelkerke $R^2$ of the model | 0.033 | | | 0.154 | | |

Notes.
Abbreviations: TSH, thyroid stimulating hormone; FT4, free thyroxine; SLE, stressful life events.

**Table 4  Results of the logistic regression exploring the relationship between thyroid hormones and clinical phenotype at onset.**

| | Criteria C cycloid psychosis | | | Criteria D cycloid psychosis | | |
|---|---|---|---|---|---|---|
| | OR | CI 95% OR | p value | OR | CI 95% OR | p value |
| TSH | 0.848 | 0.522–1.378 | 0.507 | 0.958 | 0.489–1.874 | 0.899 |
| FT4 | 2.056 | 0.225–18.787 | 0.523 | 1.900 | 0.098–36.680 | 0.671 |
| Female gender | 3.030 | 0.944–9.728 | 0.062 | 2.093 | 0.424–10.325 | 0.365 |
| Age | 0.975 | 0.933–1.018 | 0.250 | 0.955 | 0.894–1.019 | 0.166 |
| SLE at onset | 1.236 | 0.391–3.911 | 0.719 | 1.152 | 0.223–5.949 | 0.866 |
| Cannabis (positive urine test) | 0.783 | 0.219–2.804 | 0.707 | 0.329 | 0.051–2.137 | 0.244 |
| Nagelkerke $R^2$ of the model | 0.134 | | | 0.133 | | |

Notes.
Abbreviations: TSH, thyroid stimulating hormone; FT4, free thyroxine; SLE, stressful life events.

and phospholipase-C systems). In a previous cross-sectional study (*Barbero et al., 2015*) including young (aged 18–35 years) early psychotic patients (less than 3 years of illness), better cognitive performance in the attention/vigilance domain was positively correlated with FT4 levels (but not TSH or thyroid antibodies). No other previous studies have explored how thyroid function correlates with clinical phenotype in FEP patients. Most studies have focused on other clinical aspects of the illness and have shown a close relationship between thyroid abnormalities and psychosis in general (*Othman et al., 1994*). Although we did not find a significant association between thyroid hormones and the full cycloid psychosis criteria, it is important to underscore that only nine patients (9.2%) fulfilled all criteria for cycloid psychosis. Therefore, our sample might be underpowered for detecting significant associations with this outcome due to the low prevalence of cycloid psychosis. Further multicentre studies might overcome this limitation by increasing the number of patients with cycloid psychosis.

Thyroid dysfunction findings have been mostly associated with affective psychosis (bipolar disorder) rather than with nonaffective psychosis (schizophrenia) (*Carta et al., 2004*). We found that patients diagnosed with affective psychosis at 12 months showed

higher free thyroxine levels at onset than nonaffective psychosis patients. This result would be in line with that found in a previous study (*Barbero et al., 2015*), where an exploratory analysis by psychotic subtypes suggested that subjects with affective psychoses (BD or schizoaffective disorder) had increased FT4 levels and a better cognitive profile than those with nonaffective psychosis (schizophreniform disorder or schizophrenia). However, in a recent study (*Petruzzelli et al., 2020*) with child and adolescent patients, those diagnosed with first affective spectrum disorder showed lower free thyroxine levels than those with first schizophrenia spectrum disorder.

We did not find an association between the severity of psychopathology and thyroid function, which is in accordance with previous studies including patients with FEP (*Barbero et al., 2020*; *Jose et al., 2015*) or bipolar disorder (*Goyal, Yadav & Solanki, 2021*; *Barbero et al., 2014*). However, in previous studies including patients with schizophrenia, the severity of illness showed a positive correlation with thyroxine levels (T4) (*Baumgartner, Pietzcker & Gaebel, 2000*) and a negative correlation with free T3 levels (*Ichioka et al., 2012*).

Tetrahydrocannabinol, the main psychoactive constituent present in cannabis, acutely alters several hormones, including suppression of luteinizing hormone (*Cone et al., 1986*), testosterone (*Barnett, Chiang & Licko, 1983*), and triiodothyronine (T3) (*Parshad, Kumar & Melville, 1983*). Cannabinoids also suppress the hypothalamic–pituitary–adrenal (HPA) axis at the pituitary level and thyroid gland (*Chakrabarti, 2011*). Despite this, there are limited human data regarding the effects of cannabis on thyroid hormone levels. Two previous studies found lower TSH and T3 levels in acute cannabis users (*Malhotra et al., 2017*; *Parshad, Kumar & Melville, 1983*), but these changes were not seen in chronic users (*Bonnet, 2012*) or in cannabis-related psychosis patients (*Muzaffar et al., 2021*). In our study, we found that FEP patients with a positive cannabis test in urine at admission had higher FT4 concentrations, although they maintained similar TSH concentrations than FEP patients with a negative cannabis test. To our knowledge, this is the first time these results have been found in a sample of FEP patients.

Some limitations of our study need to be addressed. First, we only determined thyroid function once (one morning sample collected under fasting conditions). Second, we did not assess thyroid autoimmune status, which might be associated with the clinical phenotype of FEP (*Barbero et al., 2020*). Third, triiodothyronine (T3) was not assessed in our study because this hormone is not usually determined in routine clinical practice. Lastly, as already mentioned, the sample size of our study might be too small for detecting associations between thyroid hormones and low-prevalence phenotypes (*e.g.*, cycloid psychosis).

In summary, our study suggests that higher free-thyroxine levels are associated with a specific clinical phenotype of FEP patients (fewer prodromal symptoms, shorter DUP duration and sudden onset of psychosis) and with affective psychosis diagnoses at the 12-month follow-up. These findings highlight the need to monitor thyroid function in specific psychotic phenotypes (*e.g.*, sudden onset of psychosis) in order to detect potential thyroid abnormalities and treat them if necessary.

## ACKNOWLEDGEMENTS

The authors thank the Adult and the Child and Adolescent Psychiatry Services of Hospital de Mataró for their care of patients and collaboration with the study. We are indebted to the patients and their families who participated in the study.

### Funding

The authors received no funding for this work.

### Competing Interests

The authors declare there are no competing interests.

### Author Contributions

- Eloi Gine-Serven conceived and designed the experiments, performed the experiments, analyzed the data, prepared figures and/or tables, authored or reviewed drafts of the article, and approved the final draft.
- Maria Martinez-Ramirez performed the experiments, analyzed the data, prepared figures and/or tables, authored or reviewed drafts of the article, and approved the final draft.
- Ester Boix-Quintana performed the experiments, analyzed the data, prepared figures and/or tables, authored or reviewed drafts of the article, and approved the final draft.
- Eva Davi-Loscos performed the experiments, analyzed the data, prepared figures and/or tables, authored or reviewed drafts of the article, and approved the final draft.
- Nicolau Guanyabens performed the experiments, analyzed the data, authored or reviewed drafts of the article, and approved the final draft.
- Virginia Casado performed the experiments, analyzed the data, authored or reviewed drafts of the article, and approved the final draft.
- Desiree Muriana performed the experiments, analyzed the data, authored or reviewed drafts of the article, and approved the final draft.
- Cristina Torres-Rivas performed the experiments, analyzed the data, authored or reviewed drafts of the article, and approved the final draft.
- M. J. Cuesta analyzed the data, prepared figures and/or tables, authored or reviewed drafts of the article, and approved the final draft.
- Javier Labad conceived and designed the experiments, prepared figures and/or tables, authored or reviewed drafts of the article, and approved the final draft.

### Human Ethics

The following information was supplied relating to ethical approvals (i.e., approving body and any reference numbers):

The Hospital de Mataro granted Ethical approval to carry out the study within its facilities (CEIC 1/18).

## Data Availability

The raw data are available in the Supplemental Files.

## Supplemental Information

Supplemental information for this article can be found online at http://dx.doi.org/10.7717/peerj.15347#supplemental-information.

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
