# Peer review of "Association between free thyroxine levels and clinical phenotype in first-episode psychosis: a prospective observational study"

_PeerJ, doi:10.7717/peerj.15347_

## Round 0.1 · original submission · Major Revisions

We have received reviews from the reviewers for your manuscript. This is a well-written article; however, some areas need improvements. Please include a paragraph in the Discussion section – with regard to the implications of the findings for clinical audiences.

Reviewer 1 ·

Basic reporting

1. The manuscript is structured in a logical manner and written in professional English. Minor modifications for written English are below:

Line 32: Change "Spearman correlation was conducted" to "Partial correlation analysis was conducted"

Line 40: change “free thyroxine levels” to “FT4 levels”

Line 56: Change "psychosis, however the overwhelming" to "psychosis; however, the overwhelming"

Line 145: change "determined by chemiluminescence (ECLIA)" to "determined by electrochemiluminescence immunoassay (ECLIA)

Line 146: Change "Free thyroxin (fT4)" to "Free thyroxine (FT4)"

Line 180: change "1,21" to "1.21"

Line 207: change “free T4 levels” to “FT4 levels”

2. Even though the authors provide the summary statistics for the patient data in Table 1, it would be beneficial to provide the raw data for the audience and to ensure the reproducibility and credibility of the study. Therefore, I recommend that the authors submit raw patient data that were used for the statistical analysis.

3. The rationale for the study was clearly stated and built on the past studies. In addition, the authors clearly stated the limitations of the study at the end of discussion, providing a balanced, realistic view on the significance of the study. This is helpful for the audience and provides additional credibility to the study overall, and I commend the authors for this contribution.

4. To explain the observed association of FT4 levels and clinical phenotypes of FEP, the authors provided additional biological insights to explain the findings in discussion. This was helpful and strengthens the study rather than simply reporting the observed correlation.

Experimental design

5. The research question is well-defined, and the purpose of the study is clear and founded on previous studies.

6. Please provide more details on how the thyroid functional studies were performed. For instance, what instrument was used for ECLIA? Did the authors use any kit to measure the thyroid hormones? Which antibody was used? To ensure reproducibility and cross-comparison with other studies, it is critical to clarify how the thyroid levels were measured in the study.

7. Could you please comment on why triiodothyronine (T3) levels were not measured in the study? T3 is an integral part of the thyroid hormone biology, whose release is also stimulated by TSH like T4, and based on the findings from the study, it is highly possible that T3 levels are associated with the observed clinical phenotypes.

8. I commend the authors for the proper selection of statistical methods used in the study. Partial correlation analysis, as opposed to simple correlation analysis, was appropriate due to the existence of the confounding variables in the study, and Spearman correlation is likely more appropriate than Pearson correlation in clinical data like this. The choice of logistic regression analysis was also appropriate.

Validity of the findings

9. The results from partial correlation analysis and logistic regression analysis were reported to conclude that:

• Patients with prodromal symptomatology showed lower baseline FT4 levels

• The duration of DUP was inversely associated with FT4 levels. FEP patients with sudden onset of psychotic symptoms showed higher FT4 levels at admission

• Patients diagnosed with affective psychotic disorders at the subsequent 12-month follow-up showed higher FT4 levels at admission than patients diagnosed with nonaffective psychosis

Overall, the reporting of the statistical analysis could be improved, and I suggest the authors the following items to strengthen the validity of the statistical findings.

• Please report 95% confidence intervals for each odds ratio. This is a standard for reporting logistic regression analysis results, and it would help the audience interpret the results better

• As stated in point 2, please provide the raw patient data used for the study.

• For logistic regression, what was the Akaike information criterion (AIC) of the model? AIC assesses the goodness of fit of the model, and reporting AIC is a standard and would also help the audience interpret the findings better.

Additional comments

Overall, the study was conducted with a scientifically and logically sound hypothesis, and appropriate statistical methods were used. I suggest the manuscript be accepted with minor modifications mentioned above.

Annotated reviews are not available for download in order to protect the identity of reviewers who chose to remain anonymous.

·

Basic reporting

Introduction: Didn't follow the funnel shape pattern of introduction. Add the global, country, and state wise prevalence of of FEP.

Experimental design

Provide IEC number.

Validity of the findings

No comment

Additional comments

Discussion: add the latest findings of articles. rewrite the discussion.

·

Basic reporting

Slight modifications are needed in Table 1 & 2

Experimental design

no comment

Validity of the findings

Some minor changes needs to be done. Please see the additional comments

Additional comments

What is the level of agreement between the two trained psychiatrists who carried out diagnostic interviews
How do you adjust for some variables in correlation? Adjusting is possible only in regression
In Table 1: number of observations for each variable should be designated with “n”, not “N”
In Table 2: Keep the categories of exposure variables and mention the OR for each category and mentioning “Reference” for reference category.

---

## Round 0.2 · accepted · Accept

The authors have addressed the reviewers' comments, the manuscript may be accepted as per journal policy.

Reviewer 1 ·

Basic reporting

I thank the authors for addressing each of my review points with sufficient details. The revised manuscript contains details for how the statistical analyses were performed, and this will help the audience in the future.

I was not able to find the file for table S2, which contains the raw patient data that was used for the statistical analysis. This might be due to PeerJ, but please make sure to include this data in the final manuscript.

Experimental design

Logically sound.

Validity of the findings

After the revision, I support the authors' conclusions.

Additional comments

I suggest the manuscript be accepted.

·

Basic reporting

No Comment

Experimental design

No comment

Validity of the findings

No Comment

Additional comments

No Comment